# Preparation, Characteristics, and Application of Bifunctional TiO_2_ Sheets

**DOI:** 10.3390/ma13071615

**Published:** 2020-04-01

**Authors:** Xiaoyan Guo, Jiaqi Liu, Lixia Liu, Suohe Yang, Guangxiang He, Haibo Jin

**Affiliations:** 1College of Chemical Engineering, Beijing Institute of Petrochemical Technology, Beijing 102617, China; guoxiaoyan@bipt.edu.cn (X.G.); 2018520011@bipt.edu.cn (J.L.); liulixia@bipt.edu.cn (L.L.); yangsuohe@bipt.edu.cn (S.Y.); hgx@bipt.edu.cn (G.H.); 2Beijing Key Laboratory of Fuels Cleaning and Advanced Catalytic Emission Reduction Technology, Beijing 102617, China

**Keywords:** TiO_2_, cellulose acetate, bifunctional, sheets, dry reagent

## Abstract

TiO_2_ is a high-reflectance material for preparing sheets during dry reagent chemical tests in detail. In this study, bifunctional TiO_2_ sheets with diffusive and reflective properties were prepared using TiO_2_ microspheres (particle size 2–3 µm) and cellulose acetate (CA). Factors such as the CA dosage, water content, mixing time, and the choice of surfactant were investigated. The structure and properties of the bifunctional TiO_2_ sheets were characterized by thermogravimetry and differential thermal analysis (TG-DAT), scanning electron microscopy (SEM), dynamic contact angle test and reflectance spectroscopy. By studying the above experimental results, it was concluded that the most optimal preparation conditions for preparing the bi-functional TiO_2_ sheets under natural drying conditions were as follows: the mass ratio of CA to TiO_2_ microspheres was 0.05:1; Triton-100 was used to improve the diffusion performance of the bifunctional sheets, after mixing for 5 h and coating. The light reflectivity of the bifunctional TiO_2_ sheets in the 420 to 800 nm range was higher than 90%. Serum diffused in the bifunctional TiO_2_ sheets reacted in the reagent sheets and formed uniform colorful spots. Considering the repeatability of spot proportion and light reflectivity, the sheet offered a uniform serum diffusion and good repeatability. So, the bifunctional TiO_2_ sheets are nominated as a promising material for dry chemical diagnostic reagents.

## 1. Introduction

Dry reagent chemical tests, which have advantages such as long stability, decreased pollution, and easy operation compared to liquid reagents, are widely applied in clinical and food analysis. The quantification of these tests is typically carried out by using reflectance spectroscopy [1]. High-reflectance materials such as titanium dioxide, barium sulfate, zinc oxide, and lead oxide can be used to prepare reflective sheets. Porous diffusion sheets are absolutely necessary to uniformly distribute the test solution. While some commercial filter papers have been used as a diffusion sheet, they increase the test solution volume because of their water-absorbing capacity.

Of the high-reflectance materials mentioned above, TiO_2_ has been extensively studied because of its great potential in several fields: photocatalysis [2,3], energy storage and conversion [4,5,6,7,8], and sensors [9,10,11,12,13,14]. Moreover, it has antifouling properties [15,16], antibacterial activity [17,18], and can be used for filters [19] and pigments [15,20,21]. Porous TiO_2_ sheets are prepared using a papermaking technique which blends ceramic fiber with TiO_2_ [22]. Both TiO_2_ and S-doped TiO_2_ (S-TiO_2_) are immobilized on flexible low-cost aluminum sheets using a simple sol-gel dipping process [23].

Some researchers have investigated the use of TiO_2_ blends in organic polymer membranes to increase its hydrophobicity. Those membranes prepared using an easy-operation and quick-preparation method have good performance on permeate flux in filtration. A study on cellulose acetate/TiO_2_ hybrid membranes reported that the addition of TiO_2_ nanoparticles leads to an enhanced permeate flux of water [24]. Another study on cellulose acetate butyrate/TiO_2_ hybrid membranes reported that the addition of TiO_2_ enhanced the rejection and permeate flux infiltration of bovine serum albumin solution [25]. Those membranes are modified organic polymer membranes with TiO_2_ concentration varying from 0 to 25% [24]. However, few previous studies have been conducted to increase TiO_2_ concentration over 95% even to prepare TiO_2_ sheets while using less than 5% cellulose acetate as an organic polymer.

In this study, TiO_2_ microspheres and a cellulose acetate solution were blended to prepare bifunctional TiO_2_ sheets with diffusive and reflective properties. The properties and performance of the prepared sheets during dry reagent chemical tests were investigated in detail. The structure and properties of the TiO_2_ sheet were analyzed using thermogravimetry and differential thermal analysis, scanning electron microscopy, dynamic contact angle test and reflectance spectroscopy.

## 2. Materials and Methods

### 2.1. Materials

2–3 µm TiO_2_ microspheres were prepared in our own lab [26]. Cellulose acetate with an average molecular weight of 30,000 g·mol^−1^ was procured from Sino pharm Group (Beijing, China). Triton-100 (J&K Scientific, Beijing, China) was used as the surfactant. Acetone and dehydrated ethanol were procured from the Beijing reagent factory (Beijing, China), which were used as the solvent. Distilled water was used as a non-solvent. Serum was donated by Ortho-clinical Diagnostics, Inc (Rochester, NY, USA).

### 2.2. Sheet Preparation

Cellulose acetate (CA) solution was prepared from 1 g CA, and 12 mL acetone. Subsequently 1 g TiO_2_ microspheres, 5.26 µL Triton-100, and CA solution varied mass fraction from 0.04 to 0.065 were added into the mixture. The mixture was agitated for 3 h at 350 r/min and kept at 25 °C for 5 h to remove the air bubbles.

Afterward, 300 µm thick wet sheets were cast on a glass plate using a film applicator. The cast sheets were subsequently steamed and then immersed in a 25 °C distilled water bath. Finally, the membranes were heat-treated in a 50 °C deionized water bath for 20 min to remove the excess acetone.

### 2.3. Sheet Characterization

#### 2.3.1. Scanning Electron Microscopy

The top surface and cross-section of the sheets were observed with a ZEISS SUPRA55 scanning electron microscope (SUPRA-55 SEM, Zeiss, Jena, Germany) [26]. The sheets were cut into small pieces under liquid nitrogen to obtain a generally consistent and clean cut. The sheets were sputter-coated with a thin gold film and then mounted on brass plates with double-sided adhesive tape. Photomicrographs were taken in very high vacuum conditions at 5 kV.

#### 2.3.2. Dynamic Contact Angle Test

The dynamic contact angle of the sheets was tested by an Attension C-201 (Attension C-201, Biolin Scientific, Gothenburg, Sweden) optical contact angle tester [27]. The specific operation steps were as follows: i) the sample was fixed on the test platform; ii) the droplets were dripped from the needle of the syringe onto the surface of the sample, and the dynamic value of the contact angle was read out by the instrument through rapid photography. The liquid used in the contact angle measurement was deionized water, and the amount of water in the needle was 3 µL when measured.

#### 2.3.3. Thermal Properties

Thermal degradation was conducted using a thermal gravimetric analyzer (SDT-Q600, TA, New Castle, PA, USA). A 25 mg sample was loaded in a pretarred platinum pan and preheated it above 120 °C to remove the moisture. After cooling to 25 °C, the sample was reheated to 700 °C at a 20 °C/min rate.

#### 2.3.4. Water Content

The wet weights of the sheets were obtained after soaking the sheets in water for 24 h. The sheets were weighed after wiping with blotting paper. The wet sheets were placed in a vacuum drier at 75 °C for 48 h and the dry weights of the sheets were determined [12,22]. The percentage of water content (WC) was calculated using the following Equation (1):(1)$$\mathrm{WC}\%=\frac{W_{\mathrm{wet}}-W_{\mathrm{dry}}}{W_{\mathrm{wet}}}\times 100\%$$

W_wet_ and W_dry_ are the wet and dry weights of the sheets, respectively.

#### 2.3.5. Reflection Tests

Reflection tests were performed using a UV-Vis-NIR spectrophotometer (Lambda 950, Perkin Elmer, Waltham, MA, USA) with an integral sphere (Lab sphere, 150 mm RSA ASSY) [26].

#### 2.3.6. Diffusion Performance Tests

The reagent sheet was prepared and cut into 6 pieces, then a TiO_2_ bifunctional sheet was prepared on each reagent sheet. Six 10 µL serum samples with 600 mg/dL of glucose were added onto the surface of those TiO_2_ bifunctional sheets, separately, which were then batch operated at 37 °C for 10 min. When the serum diffused through the sheets and reacted in the reagent sheets, colored spots formed. The diameters of the spots and the reflection rate were determined.

The standard deviation (SD) was calculated using the following Equations (2) and (3):(2)$$\mathrm{SD}\%=\sqrt{\frac{\sum_{i=1}^{n}{\left(X_{i}-\bar{X}\right)}^{2}}{n}}\times 100\%$$
(3)$$\bar{X}=\frac{\sum_{i=1}^{n}X_{i}}{n}$$

## 3. Results

### 3.1. Analysis of TiO_2_ Particles

The result of the particle size analyzer evaluation of the TiO_2_ micro-particles is shown in Figure 1. As this figure shows, the average particle size of TiO_2_ was equal to 2.37 µm.

### 3.2. Sheet Morphology

Bifunctional sheets were prepared according to the above methods. When a low mass ratio of CA to TiO_2_ (0.04:1) was used, it was unable to prepare uniform sheets and the sheets broke to pieces easily. As the mass ratio increased to 0.045:1, the adhesion degree between CA and TiO_2_ improves, but the sheets were still easy to crack. Using higher CA to TiO_2_ mass ratios of 0.05:1, 0.06:1, and 0.065:1, the sheets formed a uniform, white, shiny monolithic laminar material as shown in Figure 2.

SEM images were used to observe the porous surface morphology and determine the effects of a CA to TiO_2_ mass ratio varying from 0.05:1 to 0.06:1. The results are shown in Figure 3.

With a mass ratio of 0.05:1, the pore distribution of the TiO_2_ sheets was dense and easy to present a uniform aperture size (Figure 3a). When the mass ratio increased to 0.055:1, the aperture size decreased because the TiO_2_ microspheres were coated with CA. (Figure 3b). As the CA to TiO_2_ mass ratio increased to 0.06:1, more TiO_2_ microspheres were wrapped or more pores were filled with CA (Figure 3c). In the preparation process, acetone solvent is used to completely dissolve CA, and TiO_2_ can be combined more evenly.

The different morphologies and properties of the TiO_2_ bifunctional sheets were mainly caused by the different mass ratios of CA to TiO_2_. On the one hand, the effect of CA mainly showed adhesive characteristics in the system. With the mass ratio under 0.05:1, the sheets were easy to break into pieces because of poor adhesive strength with the TiO_2_ microspheres; on the other hand, the excess amount of CA can wrap the TiO_2_ microspheres or fill some pores which is not conducive to the diffusivity of the solution.

### 3.3. Contact Angle Test

Contact angle dynamic continuous test analysis was performed on the sheet. When the droplets dropped, the surface of the sheet was wetted by the liquid. Then the apparent contact angle was measured from the composite surface composed of the fluid and the solid as a baseline at 3s as shown in Figure 4. It can be seen from the test results that as the mass ratio of CA to TiO_2_ increased, the diffusion of the droplets on the sheet slowed down. Therefore, with a mass ratio of 0.05:1, it can not only form uniform sheets, but also diffuse liquid rapidly.

### 3.4. Mixing Time

The optimum mass ratio of CA to TiO_2_ is 0.05:1 to prepare casting slurry. After different mixing times, the slurry solutions were cast on a coating machine using 300 µm-thick sheet applicators, which were dried naturally. Results are shown in Figure 5.

As shown in Figure 5, the TiO_2_ bifunctional sheets became smoother with longer mixing times. When mixing times were 0.5 h, 2 h, 3 h, and 4 h, separately, TiO_2_ microspheres could not disperse evenly because a part of the TiO_2_ microspheres agglomerated, which led to bulges on the sheets’ surface (Figure 5a–e). Moreover, agglomeration led to a lower amount of CA in the bulges, which makes the sheets easily broken in the area of the bulges. As the mixing time extends to 5 h, TiO_2_ microspheres evenly disperse thus the surface of sheets is smooth (Figure 5f). Consequently, to generate TiO_2_ sheets with a uniform and smooth surface, the slurry mixing time should not be less than 5 h.

To investigate the dispersion effect of the TiO_2_ bifunctional sheets, 10 µL serum was added to sheets prepared at different mixing times of 30 min, 2 h, 3 h, 4 h, and 5 h, respectively. As shown in Figure 6, after 10 s of diffusion, the serum could hardly diffuse on the surface of the 30 min-stirred sheets. The diffusion ability of the TiO_2_ sheets prepared with 2 h stirring was slightly improved. The residual liquid on the surface decreased, but the diffusion effect was still not ideal. The serum diffused fast without any residue on the surface of sheets prepared with 5 h stirring.

At short stirring times, TiO_2_ microspheres agglomerated which led to a lower amount of CA in the bulges, but a higher one in the flat areas and on the surface of the bulges. With a CA to TiO_2_ mass ratio higher than 0.50:1, diffusion was lower as described in 3.2. As time goes on, the water in the serum evaporated and a serum sheet gradually formed on the surface of the TiO_2_ bifunctional sheets, which was more difficult to diffuse to reaction sheets. Therefore, the mixing time has a crucial influence on the diffusion effect of the serum.

### 3.5. Thermal Properties

The thermal analysis results of the CA/TiO_2_ hybrid sheets are illustrated in Figure 7. The decomposition temperature decreased as the mass ratio of CA to TiO_2_ increased because TiO_2_ has better thermal stability than CA. It can be seen that the content of CA had an influence on the thermal stability of the sheets [28]. The sheets are used at room temperature or 37 °C, so it does not affect the application of the sheets.

### 3.6. Water Content

Water content is related to the hydrophilic properties of the membrane [29]. The water content of each membrane was calculated using Equation (1). As shown in Table 1, the water content was measured for six sheets prepared under the same conditions. The results show that the average water content of the sheets was 57.1%, which indicates that the sheets are hydrophilic. The standard deviation is 0.01, which illustrates that the sheets had good repeatability at water content and the pore distribution between CA and the TiO_2_ microspheres was uniform.

### 3.7. Reflection Tests

The reflectance of the TiO_2_ sheets was detected using a UV-Vis-NIR spectrophotometer and the results are shown in Figure 8.

Figure 8 shows that the light reflectivity of TiO_2_ bifunctional sheets at 420 to 800 nm is higher than 90%. When CA/TiO_2_ (m/m) was 0.05:1, the light reflectivity of the prepared TiO_2_ sheets reached more than 90%. Especially in the wavelength range of 540 nm (glucose detection wavelength) to 670 nm, (uric acid detection wavelength), light reflectivity even reached 94.2% and 95.7%, respectively. The light reflectivity values for a group of five samples with different amounts of CA added were all ideal; thus, small amounts of CA did not significantly affect the shading function of the TiO_2_ sheets. Therefore, TiO_2_ sheets with superior light-reflective properties can be used with in vitro diagnostic dry chemical reagents.

### 3.8. Chip Detection

The standard curve of the glucose detection chip at different glucose concentrations was shown in Figure 9. With increased concentration, the color reaction between glucose and detection chip was more obvious; the color gradually deepened, resulting in the corresponding light absorption value increasing. The signal value measures the light reflection, so the corresponding signal value was reduced.

## 4. Discussion

Generally, the serum was diffused by the TiO_2_ bifunctional sheets and reacts with the reagent sheet with a red reaction color. To explore the reproducibility of the diffusion results for TiO_2_ bifunctional sheets, the following experiments were performed.

The reagent sheet was composed of the reaction substrate enzymes. In the reagent sheet, the main reaction of glucose and reagent was as follows:(4)$${\beta -D-\mathrm{glucose}+O}_{2}{+H}_{2}O\overset{\mathrm{Glcose}\;\mathrm{oxidase}}{\to }{\;D-\mathrm{gucono}+H}_{2}O_{2}$$
(5)$${2H}_{2}O_{2}+4-\mathrm{Antipyrine}+1,7-\mathrm{Dihydroxynaphthalene}\overset{\mathrm{Horseragish}\;\mathrm{root}}{\to }\mathrm{Red}\;\mathrm{dye}$$

In the lab, homemade glucose dry chemical reagent sheet was covered with the TiO_2_ bifunctional sheets to achieve glucose dry chemical diagnostic reagent conditions in vitro.

The chromogenic reaction results are shown in Figure 10. The light reflection density gauge detected the reflected light signal value for six points and the results are shown in Table 2.

Figure 10 shows that the color of the reaction spots obtained on the reagent sheet were uniform, thus confirming that the serum diffused evenly. In addition, the areas of the six spots formed after the reaction was similar. The SD of the diameter of the six spots was 0.04, which illustrates that the sheets had good diffusion repeatability for the serum. Table 2 outlines the results for the dry chemical reagent color spots tested using reflected light density testing. The relative deviation of the light reflection signal value of the six groups ranged within ±0.028%, which illustrates that the dry chemical reagent has good reproducibility. Therefore, using the same reagent sheet for the same batch, the sheets had good serum diffusivity and an excellent repeatability.

The above results indicate that the homemade TiO_2_ sheets had uniform pore distribution and aperture size on the surface, as well as inside. Therefore, TiO_2_ bifunctional sheets can be further used in dry chemical diagnostic reagents tests in vitro.

## 5. Conclusions

(1) The most optimal preparation condition for the bifunctional TiO_2_ sheets under natural drying conditions are as follows: 0.05:1 CA/TiO_2_ mass ratio, 5 h mixing after coating, and natural dried.

(2) The reflection spectrometry detection results show that the reflectivity of TiO_2_ bifunctional sheets between 420 and 800 nm can reach more than 90%, which is enough to prove that the prepared sheets have a good function of reflection. The TG-DTA analysis results show that CA disperses evenly in the TiO_2_ bifunctional sheets.

(3) The bifunctional TiO_2_ sheets are applied to dry chemical in vitro diagnostic reagents prepared using dry glucose films. The test results confirm that the spot color is uniform when the bifunctional sheets are used for the same serum concentration. The SD for the diameter of six spots is 0.04, which illustrates that the sheets have good diffusion repeatability for the serum. The relative deviation of the light reflection signal value in the six groups range within ±0.028%, which illustrates that the dry chemical reagent had a very good reproducibility. Therefore, from the preliminary results obtained, bifunctional TiO_2_ sheets are ideal to be applied to dry chemical in vitro diagnostic reagents.

## Figures and Tables

**Figure 1 materials-13-01615-f001:**
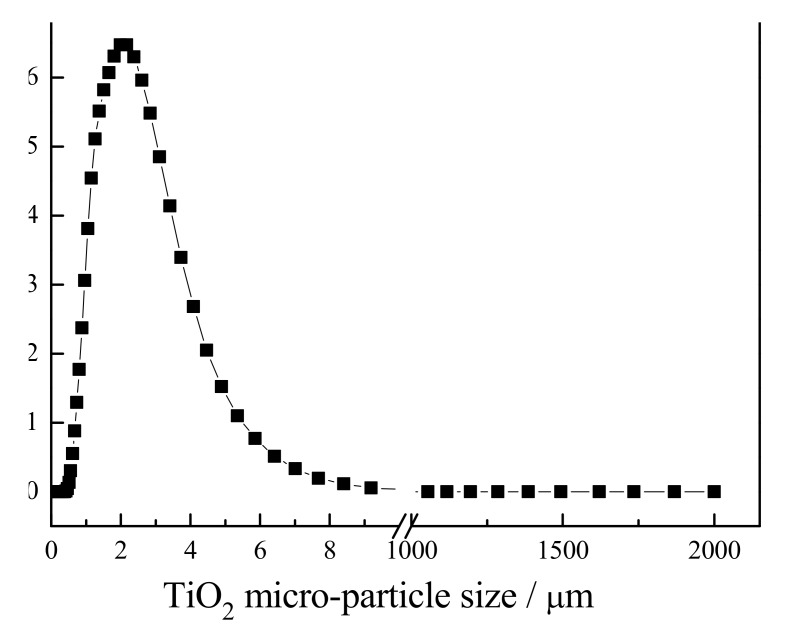
Particle size distribution for TiO_2_ micro-particle.

**Figure 2 materials-13-01615-f002:**
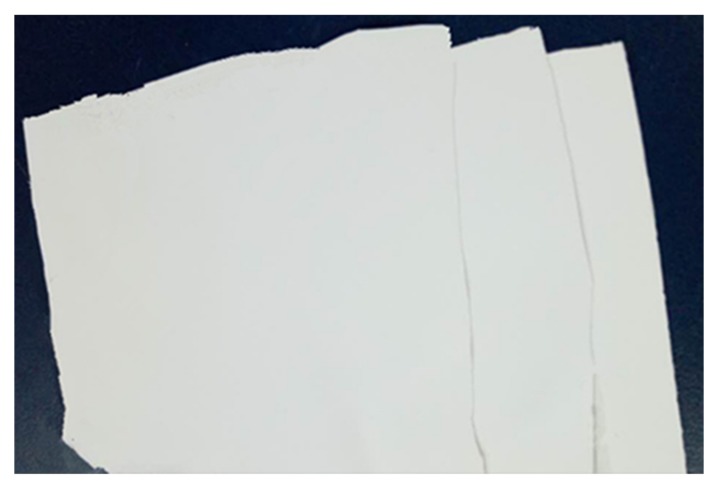
Panorama image of TiO_2_ bifunctional sheets.

**Figure 3 materials-13-01615-f003:**
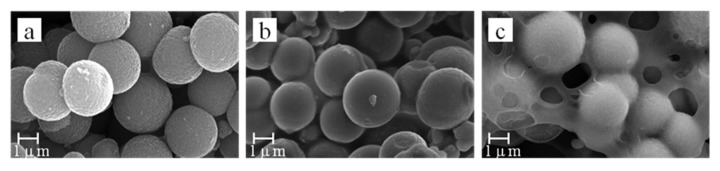
SEM micrographs of bifunctional TiO_2_ sheets with different mass ratios of cellulose acetate (CA) and TiO_2_: (**a**) 0.05:1; (**b**) 0.055:1; (**c**) 0.06:1.

**Figure 4 materials-13-01615-f004:**
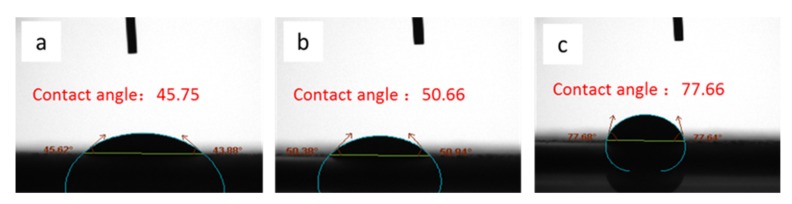
Dynamic contact angle test of bifunctional TiO_2_ sheets with different mass ratio of CA and TiO_2_ (**a**) 0.05:1; (**b**) 0.055:1; (**c**) 0.06:1.

**Figure 5 materials-13-01615-f005:**
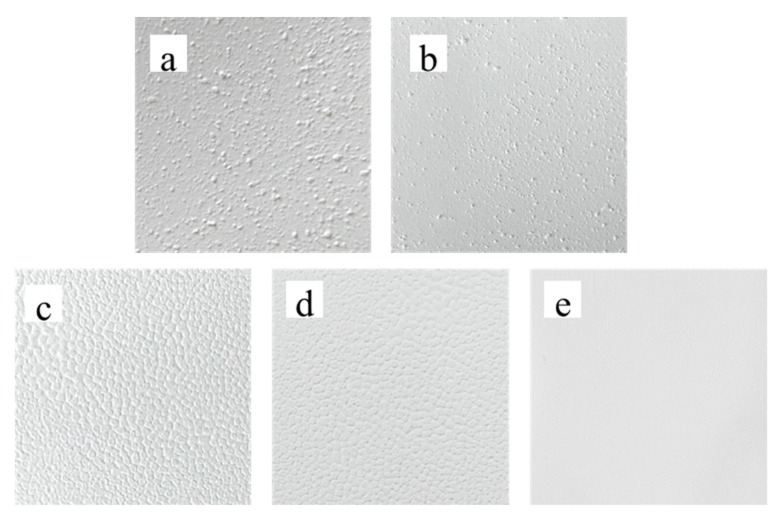
TiO_2_ bifunctional sheets prepared at different times: (**a**) 30 min; (**b**) 2 h; (**c**) 3 h; (**d**) 4 h; (**e**) 5 h.

**Figure 6 materials-13-01615-f006:**
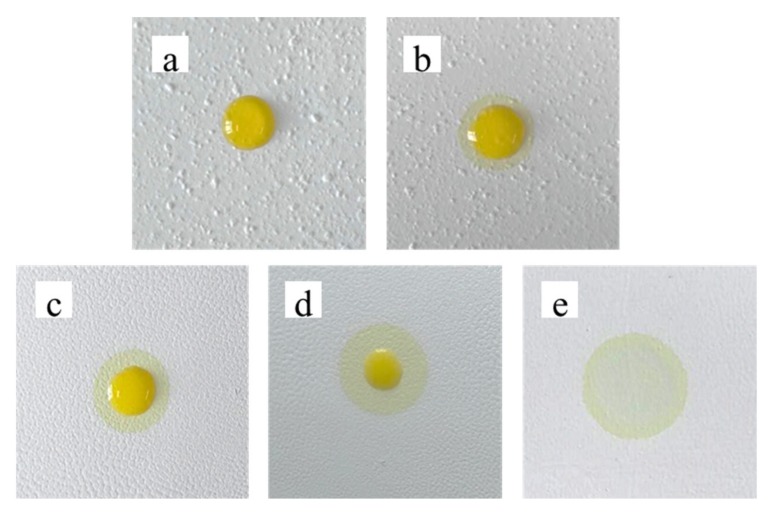
The diffusion of serum by TiO_2_ bifunctional sheets: (**a**) 30 min; (**b**) 2 h; (**c**) 3 h; (**d**) 4 h; (**e**) 5 h.

**Figure 7 materials-13-01615-f007:**
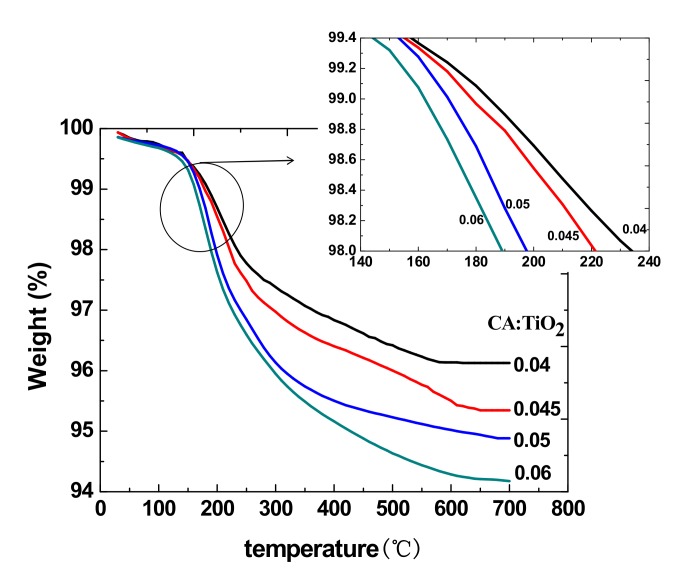
DTA-TG characterization of CA/TiO_2_ hybrid sheets.

**Figure 8 materials-13-01615-f008:**
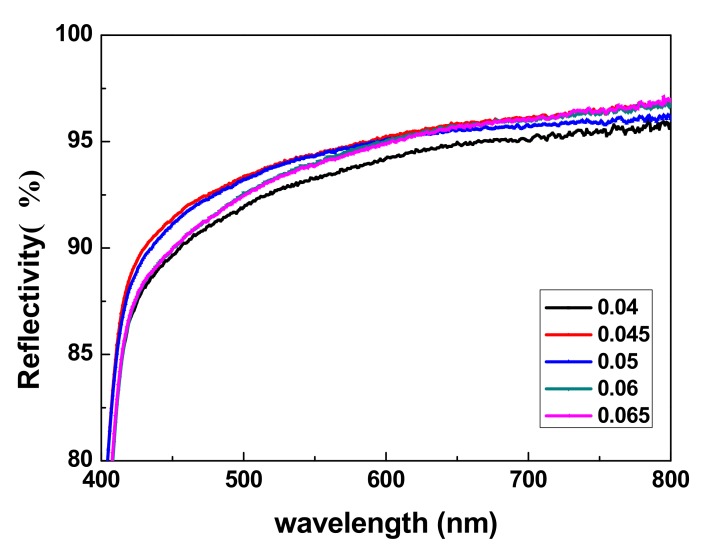
Light reflectivity of TiO_2_ bifunctional sheet.

**Figure 9 materials-13-01615-f009:**
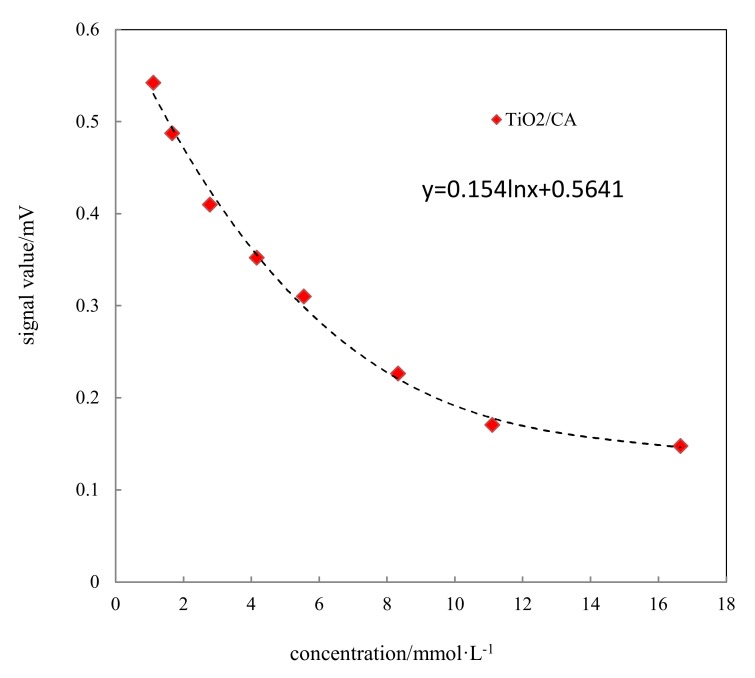
Standard curves of glucose detection chips.

**Figure 10 materials-13-01615-f010:**
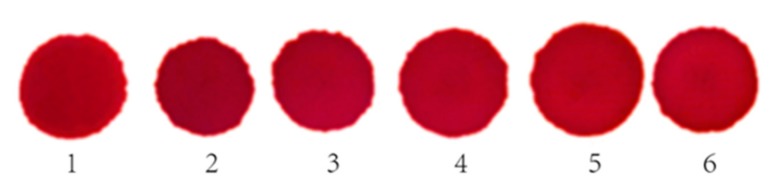
Spot images of serum reaction.

**Table 1 materials-13-01615-t001:** Different water content of each sheet.

Sheets Code	W_dry_ (g)	W_wet_ (g)	WC (%)
S1	0.025	0.059	57.6%
S2	0.026	0.061	57.4%
S3	0.026	0.062	58.1%
S4	0.026	0.060	56.7%
S5	0.026	0.059	55.9%
S6	0.026	0.060	56.7%

**Table 2 materials-13-01615-t002:** Light reflection signal values of dry chemical diagnostic reagents in vitro.

Serial	Single Value (mV)	Mean Single Value	Deviation
1	88.32	88.31	0.017
2	88.32		0.017
3	88.33		0.028
4	88.28		−0.028
5	88.28		−0.028
6	88.30		−0.006

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
