# Peer review of "Preparation, Characteristics, and Application of Bifunctional TiO2 Sheets"

_materials, 2020, doi:10.3390/ma13071615_

Round 1
Reviewer 1 Report
The work claims at investigating syntheses of Bi-functional TiO2 sheets with diffusive and reflective properties by using TiO2 microspheres with particle size 2–3 μm and cellulose acetate. The work also provides microstructural characterization of the nanostructures so-prepared.
The work is interesting and well conducted.
I don’t quite like, however, the abstract structure presented by the authors. I don’t think that so-explicit and detailed numbered description is necessary, in my opinion. Therefore, I suggest to revise the abstract not including numbered lists and concatenating the background, methods and findings in a more scientific writing, making sure that readers' attention is captured.
The introduction properly addresses the existing research and background about the topic, except in lines 50-52:
But few previous studies have been conducted to increase TiO2 concentration over 25% even to prepare TiO2 sheets using a little cellulose acetate as an organic polymer.
For these lines, I would recommend including clear background on that statement, and explain what “little cellulose” does mean in clear figures.
Methods:
Line 70: “Afterward, 300 µm thick sheets”… how can you assure that the sheet thickness is that number?
Results:
DTA-TG characterization of CA/TiO2 hybrid sheets, as shown in Figure 7, looks a bit strange, from my experience with DTA.TG… I am quite surprise that the graph does not show some typical weight losses when heating TiO2.. Please explain.
Reviewer 2 Report
Peer review of the Manuscript materials-751428 "Preparation, Characteristic and Application of Bi-Functional TiO 2 Sheets".
The manuscript needs a major review, the conclusions seems to not agree with previous sections, moreover they refer to samples which are not fully described (mixing time, CA ecc.)
p.1, abstract: may be better remove the numbered list, an abstract us a summary, not a list.
p.1 line 30: is widely -> are widely
p.1 line 34: review the sentence "While some commercial filter papers have been used as a diffusion sheet, they increase the test solution volume because of their water-absorbing capacity."
p.1 line 42 review the sentence "Both methods mentioned are complicated to perform." argumenting it.
p.2 line 76 about ZEISS SUPRA55, a reference to a datasheet should be given
p.2 line 82 sbout attension C-201, a reference to a datasheet should be given
p. 2 lines 83-87: "were as follows: The sample was fixed on the test platform. Then
the droplets were dripped from the needle of the syringe on to the surface of the sample, and the dynamic value of the contact angle was read out by the instrument through rapid photography. The liquid used in the contact angle measurement was deionized water, and the amount of water in the needle was 3 μL when measured." a point or numbered list could be useful, e.g. " were as follows: i) ...; ii) ..."
p.3 lines 99-100, for the adopted instruments please add a reference
Fig. 6: could be intersesting to add all the cases as done in Fig. 5
p. 7 line 187 and in the following "As shown in Table 1, the water content was measured
for six sheets prepared under the same conditions." which are these conditions? WC ecc. are related to foils in which conditions? 5h, 3h or something else?
Fig. 9 never called and explained in the text
Tavle 2 is not clear to me why one column has a unique value
Conclusions, point 1 "3 h mixing after coating,", at line 162 you wrote "mixing time should not be less than 5 h", what is the correct sentence?
Conclusions point 3: "range within ±0.029%," in table 2 was \pm 0.028%
Round 2
Reviewer 2 Report
The paper in this form is ok